# Characterization of RNA content in individual phase-separated coacervate microdroplets

Damian Wollny [1,2,3✉], Benjamin Vernot[1], Jie Wang[4], Maria Hondele [5,11], Aram Safrastyan [2,3], Franziska Aron[2,3], Julia Micheel [2,3], Zhisong He [6], Anthony Hyman [4], Karsten Weis [5], J. Gray Camp [7,8], T.-Y. Dora Tang [4,9✉] & Barbara Treutlein [1,10✉]

Condensates formed by complex coacervation are hypothesized to have played a crucial part during the origin-of-life. In living cells, condensation organizes biomolecules into a wide range of membraneless compartments. Although RNA is a key component of biological condensates and the central component of the RNA world hypothesis, little is known about what determines RNA accumulation in condensates and to which extend single condensates differ in their RNA composition. To address this, we developed an approach to read the RNA content from single synthetic and protein-based condensates using high-throughput sequencing. We find that certain RNAs efficiently accumulate in condensates. These RNAs are strongly enriched in sequence motifs which show high sequence similarity to short interspersed elements (SINEs). We observe similar results for protein-derived condensates, demonstrating applicability across different in vitro reconstituted membraneless organelles. Thus, our results provide a new inroad to explore the RNA content of phase-separated droplets at single condensate resolution.

[1] Max Planck Institute for Evolutionary Anthropology, Leipzig, Germany. [2] RNA Bioinformatics and High Throughput Analysis, Friedrich Schiller University, Jena, Germany. [3] Leibniz Institute on Aging—Fritz Lipmann Institute (FLI), Jena, Germany. [4] Max Planck Institute of Molecular Cell Biology and Genetics, Dresden, Germany. [5] Institute of Biochemistry, ETH Zurich, Zurich, Switzerland. [6] Biozentrum, University of Basel, Basel, Switzerland. [7] Roche Institute for Translational Bioengineering (ITB), Roche Pharma Research and Early Development, Roche Innovation Center, Basel, Switzerland. [8] University of Basel, Basel, Switzerland. [9] Cluster of Excellence Physics of Life, TU Dresden, Dresden, Germany. [10] Department of Biosystems Science and Engineering, ETH Zürich, Basel, Switzerland. [11] Present address: Biozentrum, University of Basel, Basel, Switzerland. ✉email: damian.wollny@uni-jena.de; tang@mpi-cbg.de; barbara.treutlein@bsse.ethz.ch

In the 1920's de Jong coined the term coacervation to describe a liquid-liquid phase separation process between two oppositely charged polymers in solution[1]. Electrostatic interactions bring the two components together, subsequent entropic release from water and counter ions from around the polyelectrolytes drives phase separation into membrane-free and chemically enriched micron-sized droplets[2]. These coacervate droplets have been shown to form from a wide variety of different molecules with very little chemical specificity from synthetic polyelectrolytes, to biological polyelectrolytes and small charged molecules[3]. Consequently, they were hypothesized to play a role in the origin-of-life by bringing together the first molecules to spatially localize the first primitive reactions[4]. Since then coacervates formed from synthetic polymers have been exploited in a range of industries from food separation to pharmaceuticals[5]. More recently, it has been shown that the coacervation process plays an active role in the liquid-liquid phase separation of condensates in biological systems. Whilst, the mechanism of formation of biomolecular condensates in cells has now been extensively studied, an understanding of how condensates regulate biochemical processes in time and space is still in its infancy[6,7].

Key to unraveling these unanswered questions is deconvoluting the molecular content and physicochemical properties of the condensates. So far, progress in this area has been limited by difficulty in isolating condensates from cells in their dynamic environment. To this end, in vitro reconstitution has been instrumental for in depth droplet characterization[8].

Most of the condensate characterization has relied on fluorescence microscopy. Indeed, characterization of the partition coefficients has only recently been optimized using high-throughput microfluidic methods based on fluorescence of single solutes[9]. Despite this progress, there remains no methodology to uncover the heterogenous mixture of molecules and their precise amounts in individual coacervate droplets. To this end, we have exploited single-cell RNA sequencing technology and developed a novel way to determine the amount and sequence of RNA incorporated into individual coacervate droplets. This provides an unprecedented opportunity to determine, for the first time, the RNA content of individual coacervate droplets within a population. Furthermore, we show how this method can be applied to both synthetic coacervate microdroplets and condensates prepared from biological phase separating protein scaffolds such as the human RNA-binding protein Fused in Sarcoma (FUS) and yeast DExD/H-box helicase 1 (Dhh1). We identify that RNA properties are crucial for uptake into synthetic coacervates and demonstrate comparable uptake properties to FUS and Dhh1 droplets depending on the coacervate chemical identity. Thus, our findings strengthen the role for synthetic coacervates as models for biomolecular condensates.

## Results

**Establishment of single-coacervate sequencing.** In order to determine the RNA content of individual coacervate droplets or condensates, we aimed to work with the following four droplet systems: carboxylmethyl dextran (CM-Dex) and Poly (diallyldimethylammonium chloride) (PDDA) (molar ratio: 6:1) or CM-Dex with polylysine (pLys) microdroplets (molar ratio: 6:1) in 10 mM Tris and 4 mM $MgCl_2$ at pH 8 or recombinant FUS (25 mM Tris-HCl, 150 mM KCl, 2.5% glycerol, 0.5 mM DTT, pH 7.4) or recombinant Dhh1 (50 mM KCl, 30 mM HEPES-KOH, 2 mM $MgCl_2$, pH 7.4) condensates were prepared in the presence of total RNA (50 ng/µl). The total RNA was isolated from human induced pluripotent stem cells (iPSCs) immediately before each experiment.

We started by analyzing the RNA content of CM-Dex:PDDA coacervates (Fig. 1a). The combination of the CM-Dex and PDDA polymers represent a well characterized coacervate system irrespective of the presence of RNA[9]. We first measured the partition coefficient of our RNA pool to quantify assembly of the diverse mix of RNA molecules into coacervates (mean partition coefficient = 9.46, SD = 2.08) (Supplementary Fig. 1a). The RNA-containing membrane-free droplets were then loaded into 96-well plates with each well containing 4 µl of guanidine hydrochloride (GuaHCl—6 M) by fluorescence-activated cell sorting (FACS). Using a FACS gating strategy solely based on forward (FSC) and side (SSC) scatter (Supplementary Fig. 1b), we were able to control the number of droplets sorted in each well-down to single coacervates. The presence of high concentration GuaHCl led to a change in turbidity of the coacervate dispersion from cloudy to clear which is synonymous with the dissolution of coacervate droplets (Supplementary Fig. 1c). This indicates that the coacervate droplets are dissolved upon addition to the well plate releasing the RNA from the droplets. We then purified the RNA from single coacervates using solid phase reversible immobilization (SPRI) beads[10] in order to remove the GuaHCl salt which would inhibit the subsequent reactions. Upon purification we converted RNA into cDNA and prepared Illumina sequencing libraries as previously described[11] (Supplementary Methods). Computational analysis then revealed the sequence, length and abundance of the RNA molecules which were present in each sorted coacervate. In addition, the size of the individual coacervates was obtained from FACS by the FSC.

We focused our analysis on mRNA (commonly referred to as transcripts) because of its heterogeneity in terms of sequence composition and length providing us with data from a pool of highly diverse RNAs. The abundance of each mRNA molecule present in the coacervate was quantified with the commonly used transcripts per million (TPM) metric which was particularly suitable for sample-to-sample comparisons[12]. This analysis in combination with the FACS data enabled for the first time to obtain information on both the genotype and phenotype within a population of coacervate microdroplets on a single-coacervate level.

After we successfully prepared and sequenced libraries from single coacervates we first focused on quality metrics. Bioanalyzer traces were used for quantification and quality control of the amplified cDNA from 0, 1, 10, 100 and 1000 CM-Dex:PDDA coacervate droplets (Supplementary Fig. 1d). The bioanalyzer profiles demonstrated that our approach enabled full length transcript amplification even from single coacervates. Furthermore, quantification of the amount of amplified cDNA showed a linear correlation to the amount of sorted coacervate droplets (Supplementary Fig. 1e). Wells which did not contain coacervates (negative controls), yielded only low cDNA concentration derived from primer peaks (Supplementary Fig. 1d, f). Although our FACS approach is able to reliably sort single coacervates, the liquid surrounding the coacervate within the sorted FACS droplet is likely to also contain RNA. We do, however, observe a positive correlation between the coacervate size and the number of transcripts that we sequenced from single coacervates (Supplementary Fig. 1g). This indicates that the majority of information must come from within the coacervate and not the surrounding liquid. These results demonstrate that the methodology of extraction and amplification of RNA is robust and consistent.

**Single-coacervate characterization.** Using our approach, we next aimed to investigate the relationship between coacervate size and its RNA content—specifically the relationship between the diversity of RNA transcripts, the average length of the transcripts within the coacervates and the coacervate size (Fig. 1b). Our results showed that the largest coacervates had the highest diversity of transcripts (Fig. 1c). In comparison coacervates

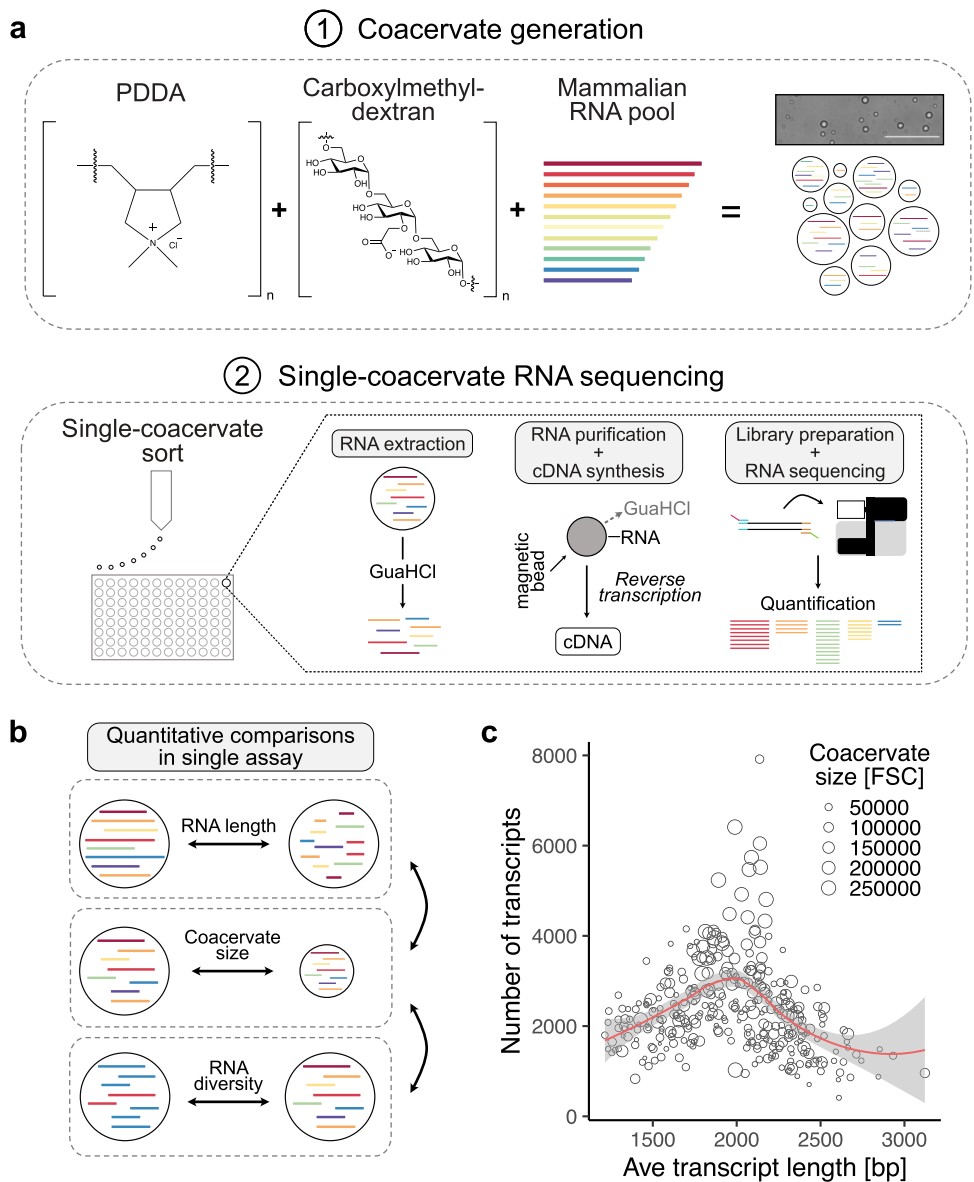

**Fig. 1 Sequencing RNA of single phase-separated coacervates. a** Schematic of coacervate generation and single-coacervate sequencing strategy. Coacervates were generated mixing charged polyelectrolytes Carboxymethyldextran with PDDA (CM-Dex:PDDA). Scale bar = 100 μm. Total RNA isolated from iPS was used as RNA input. Single coacervates were sorted into 96-well plates using fluorescence-activated cell sorting (FACS). RNA was extracted from each coacervate and mRNA was converted to cDNA and sequenced upon library preparation. RNAs present in each sequenced coacervate were computationally identified and quantified. **b** Schematic illustration of cross comparisons of several parameters (RNA length, coacervate size and complexity of RNA pool) from hundreds of individual coacervates in a single assay. **c** Relationship between the size of single coacervates, the number of different RNA transcripts and the average length of all RNA transcripts in each coacervate. Each dot represents a sequenced coacervate. Coacervate size was measured by the FACS forward scatter (FSC).

containing the longest average transcript length were among the smallest coacervates. These smaller coacervates also displayed a very low diversity of detected RNA transcripts (Fig. 1c). Interestingly, these results indicate that random pools of RNA will localize in a heterogeneous nature within dispersions of coacervate droplets leading to different phenotypic properties. Since these results only consider the pool of mRNAs, we also sequenced total RNA content from coacervates in order to probe the full diversity of RNAs within coacervates (Supplementary Fig. 2 and Supplementary Data 1). We find that the overall composition of RNA biotypes in coacervates is very similar to the RNA input. As expected, sequencing only polyadenylated transcripts misses a large number of rRNA transcripts which we do recover upon random priming (Supplementary Fig. 2). Yet, given the higher

molecular complexity of mRNAs, we decided to continue to focus on polyadenylated transcripts.

**Characterization of experiment-to-experiment variability.** Next, we wanted to test if the RNA distribution within the coacervate population was consistent across experiments. We found that the frequency with which transcripts are detected in coacervates is highly reproducible across experiments (Pearson correlation coefficient $r = 0.86$, Fig. 2a). This indicates that, albeit being a dynamic process, the localization of RNA into coacervates is not random. In contrast, how much of each transcript is present within the coacervates (TPM) is not as consistent between experiments ($r = 0.58$, Fig. 2b). This correlation remained low for both small and large droplets (Supplementary Fig. 3). Whilst

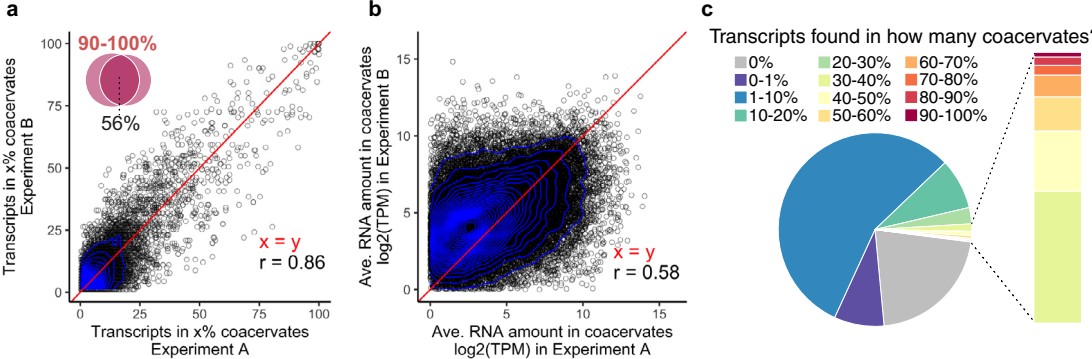

**Fig. 2 Comparison of experiment-to-experiment variability of RNA detection in coacervates. a** Quantification of the efficiency of RNA assembly into CM-Dex:PDDA coacervates across independent experiments. Each dot represents an RNA transcript. Venn diagram: overlapping transcripts across experiments that were found in 90–100% of coacervates. **b** Experiment-to-experiment variation of the average abundance of each RNA transcript across all coacervates in which it was detected. RNA abundance for each transcript is calculated as transcripts per kilobase million ($\log_2$(TPM)) enabling comparison of transcript abundances across coacervates. Red line indicates perfect correlation ($x = y$). Pearson correlation coefficient $= r$. **c** Pie chart demonstrating how frequently each input RNA transcript was detected in coacervates.

these results show that the experiments are reproducible for the type of RNA, every coacervate dispersion produced in the presence of random RNA will lead to a different heterogeneous population with respect to the amount of RNA. This has very interesting implications in considering the role of coacervation in origin-of-life and modern biological studies where each droplet within a pool may have different genotypic properties.

We further quantified how often each input RNA transcript is found in CM-Dex:PDDA coacervates (Fig. 2c). This analysis showed three things: (1) We observed that most transcripts of our input RNA pool were found in relatively few (<10%) coacervates. (2) Only 0.1% of transcripts are found in almost all (<90%) of coacervates and (3) A substantial fraction of input transcripts (21%) were not detected in any sequenced coacervates. The failure to detect these transcripts in coacervates was likely due to low abundance of these transcripts in the input (Supplementary Fig. 4), although we cannot exclude a mechanism of exclusion due to currently unknown transcript features.

**Association between RNA features and enrichment in coacervates**. Next, we investigated which RNA features determine how frequently a transcript is found in coacervates. Generally, we found a strong relationship between input amount and the frequency of detection in coacervates (Fig. 3a). We also confirmed this relationship by using synthetic RNA of different length (Supplementary Fig. 5). This indicates that the uptake of RNA is strongly dependent on the frequency of the RNA in the input. Interestingly, we found that there was a small subset of transcripts which did not follow this trend and were found in many or almost all of the coacervates, even though they were not very abundant in the input (Fig. 3a—red dots). This observation was robust across different input RNA concentrations (Supplementary Fig. 6).

We tested if other RNA features such as length or sequence might explain the efficient uptake of these transcripts. Our analysis showed that there was no correlation to the transcript length and its frequency in detection into the coacervates (Supplementary Figs. 5 and 7a, b). This result was unexpected, given that RNA length has previously been shown to be a determinant of RNA partitioning into coacervates and protein condensates[13–16]. We confirmed that the input RNA was not degraded prior to assembly into coacervates which could result in a lack of correlation (Supplementary Fig. 7c). In order to explain the discrepancy between our data and previously reported results,

we hypothesized that the effect of how abundant RNAs are in the input pool dominates over most other variables (such as length) since we saw a strong relationship between these variables in our system (Fig. 3a and Supplementary Fig. 5). We therefore analyzed the correlation between RNA length and enrichment in coacervates only for RNA molecules that are similarly abundant in the input pool (input TPM bins). We indeed found that for most bins there is a higher (and almost exclusively a positive) correlation between RNA length and enrichment into coacervates compared to when we take the whole input into consideration (Supplementary Fig. 7d).

**Sequence motif analysis of enriched RNAs**. In contrast to RNA length, sequence analysis of the RNAs which were not highly abundant in the input but were frequently found within the coacervate droplets (Fig. 3a—red dots) showed that there were sequence motifs of 11–50 bp which were enriched in the droplets compared to randomly selected non-enriched transcripts (Fig. 3b). Closer inspection of the sequence motifs revealed that the two most highly ranked motifs (Motif 1 and Motif 2) were in fact almost perfect reverse complements of each other (Fig. 3b and Supplementary Fig. 8). To investigate the effect of Motif 1 and Motif 2 on transcript uptake by coacervates, we looked at the efficiency of uptake of an RNA transcript which contained both motifs. We found that transcripts which contained both motifs on the same transcript were detected more frequently within a coacervate compared to transcripts containing just Motif 1 or Motif 2 alone (Fig. 3c). The distances between these motifs on the transcripts were, however, too large to suggest hairpin structures (Supplementary Fig. 9a), potentially pointing toward more intricate secondary RNA structure as no global folding differences are observed between enriched and random transcripts (Supplementary Fig. 9b). This is further supported by the fact that some motifs in enriched transcripts have very defined distances (median distances: 70, 71, 53, 84 bp for motifs 1, 4, 6 and 9 respectively) to each other when detected on the same transcript (Supplementary Fig. 10a, b). In contrast, all motifs found in randomly chosen transcripts displayed a broad distribution of distances to other motifs suggesting no obvious structural relationship between those motifs (Supplementary Fig. 10c, d). Since this result suggested that RNA-RNA interaction through sequence complementarity on the same transcript might be an important determinant of efficient RNA uptake into coacervates, we further investigated sequence complementarity across different transcripts. We found that the

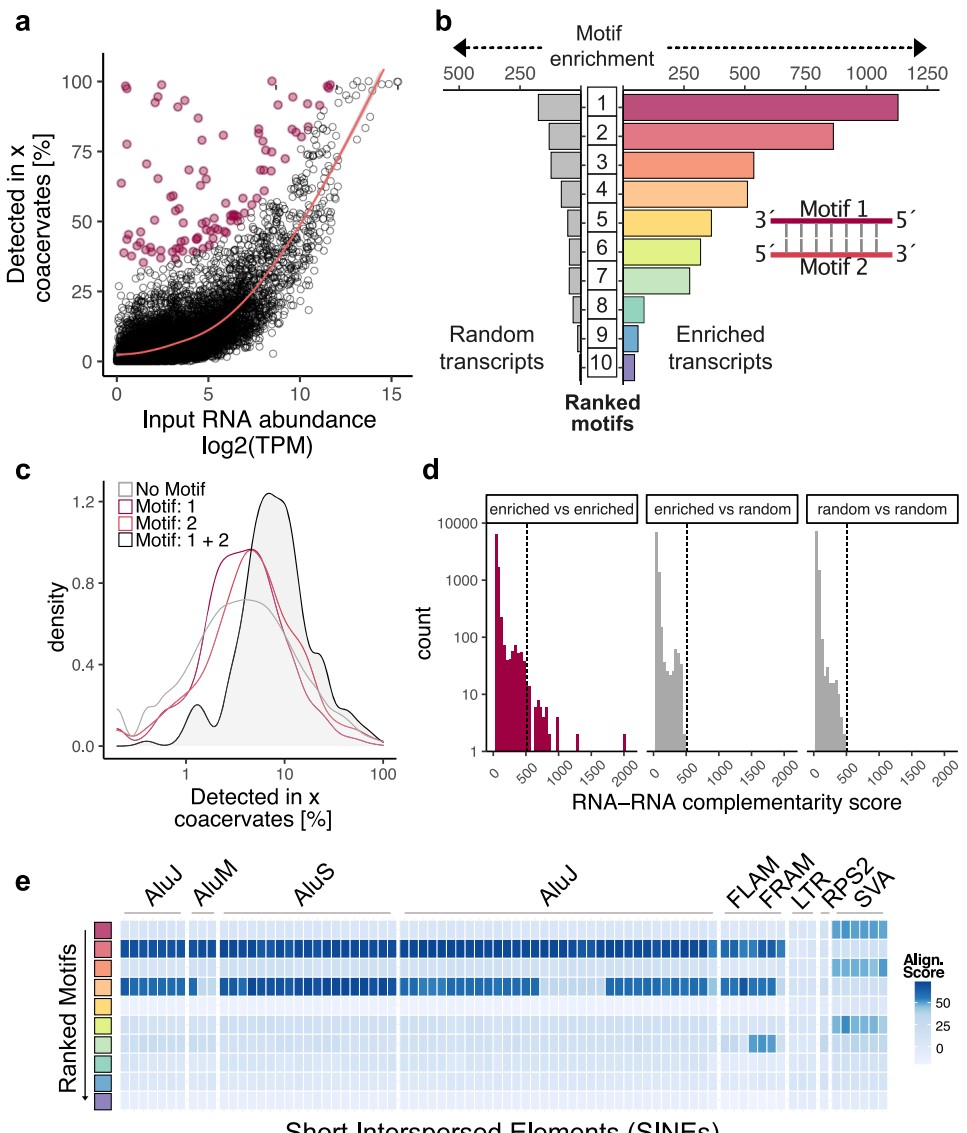

**Fig. 3 Properties of RNA found within coacervates. a** Correlation between input RNA amount and the frequency with which each transcript is detected in CM-Dex:PDDA coacervates. Transcripts that are enriched in coacervates (defined as residuals >30 for generalized additive model) are labeled red. **b** Analysis of sequence motifs that are detected within enriched transcripts (as defined in **a**) or randomly selected non-enriched transcripts. Motif enrichment corresponds to *E*-value derived from MEME suite. Among enriched transcripts, the two most abundant sequence motifs (Motifs 1 and 2) display sequence complementarity. **c** Frequency of transcript detection in coacervates conditional on if the transcripts contain either Motif 1, Motif 2, both motifs or none. **d** Analysis of sequence complementarity among different transcripts present in the pool of enriched or randomly selected transcripts. Sequence complementarity was determined using local-pairwise alignment (Smith–Waterman) scores. Dotted line indicates the maximum complementarity score that was detected outside the enriched vs. enriched comparisons (gray bars). **e** Comparison of sequence similarity of enriched motifs to known genomic elements. Heatmap represents pairwise alignment (Smith–Waterman) of enriched motifs with sequences of short interspersed elements (SINEs). Color intensity represents alignment score.

pool of enriched transcripts contains transcript pairs with very high sequence complementarity compared to enriched vs. random transcripts or random vs. random transcripts (Fig. 3d). In order to more directly test the impact of double-stranded RNA formation for uptake into coacervates we synthesized fluorescently-labeled oligonucleotides of Motif 1 and Motif 2 and quantified the uptake with flow cytometry as well as confocal microscopy. While quantifying a large number of coacervates ($n = 10,000$), we observed that coacervates take up more double-stranded RNA composed of Motif 1 and 2 (mean partition coefficient = 10.2, SD = 1.01) compared to each motif alone or scrambled motifs (Supplementary Fig. 9c, d).

Next, we sequence matched the discovered sequence motifs to match any known genomic features. The motifs showed high similarity to genomic regions annotated as short interspersed elements (SINEs). SINEs belong to the family of transposable elements which have the potential to regulate transcription or generate new transcript isoforms[17]. In order to systematically test for sequence homology, pairwise alignment of each motif with SINE family members was undertaken (Fig. 3e). It was found that two motifs (Motif 2 and 4) show strong sequence similarity to Alu elements which are primate specific transposable elements which are highly abundant in the human genome[18]. Three motifs (Motif 1, 3 and 6) display similarly high homology to hominid-specific

SINE-VNTR-Alu retrotransposons which also have an Alu element as their main component[19].

Since many of the top motifs that we discovered resemble primate-/hominid-specific elements we tested which motifs enrich in coacervates when we take a transcript input pool isolated from e.g., a murine cell line. When we performed single-coacervate sequencing with coacervates that contain RNA derived from mouse embryonic fibroblasts (MEF), we again observed that most RNAs enter coacervates at high frequency strongly depended on how abundant these RNAs were in the input with the exception of the outlier transcripts as expected from our previous results (Supplementary Fig. 11a). Upon motif enrichment analysis we did not observe any sequences that resemble SINE motifs (Supplementary Fig. 11a). Instead, we saw an enrichment of A and T stretches (Motifs 2, 3, 7) as well as G and C stretches (Motifs 1, 4, 8) (Supplementary Fig. 11b). Additionally, we found one motif (Motif 5) with high sequence complementarity to B1 SINE elements (Supplementary Fig. 11c). Overall, these results again suggest the enrichment of dsRNA structure of mouse-derived RNA in coacervates. Yet, in contrast to human RNA, the complementarity appears to be driven less by SINE elements but rather by mononucleotide repeats.

In order to test the in vivo relevance of our findings, we analyzed bulk RNA sequencing data from stress granules[15] as well as P-bodies[20] which were isolated from cells and searched for the most enriched motifs in both datasets. Interestingly, we found that the two most enriched motifs from our CM-Dex:PDDA coacervates are (among other motifs) also enriched in stress granules and in P-bodies (Supplementary Fig. 12). These results indicate that the motifs we discovered are also relevant for RNA assembly into condensates in cells, even though stress granules and p-bodies are more molecularly complex than our in vitro coacervate system.

**Sequencing diverse coacervate and condensate types**. As the single-cell sequencing methodology is applicable to both synthetic coacervate droplets and to condensates which are formed from protein scaffolds we compared the RNA accumulation properties between different systems. We generated coacervates from CM-Dex with polylysine (CM-Dex:pLys, 6:1 molar ratio) to compare the results obtained so far to another synthetic coacervate system. Lysine residues are enriched in disordered regions of P-body condensate proteins and its polymer form has been shown to form condensates which support complex enzymatic reactions[21,22]. Additionally, we sequenced RNA from well characterized Dhh1 and FUS-based phase-separated droplets in order to compare RNA accumulation in synthetic coacervates vs. protein-based condensates (Fig. 4a)[7,23].

We first looked at how often any given transcript is detected in the different droplet systems and compared all results (Fig. 4b). We found a high correlation between all condensate types in particular for PDDA and FUS condensates (Fig. 4b). These results demonstrate that many RNAs that frequently localize in droplets will do so, irrespective of the host molecules of the droplets. However, there is a subpopulation of transcripts that are taken up more efficiently in a condensate-type specific way which was not a result of differences in the input (Fig. 4b and Supplementary Fig. 13).

For global cross comparison of all sequenced condensates we performed a dimensionality reduction analysis followed by Uniform Manifold Approximation and Projection (UMAP)[24]. This analysis evaluates how comparable all profiled condensates are to each other with respect to the RNA transcripts they contain. For this analysis, we focused on input independent, enriched transcripts for each condensate type (as defined in

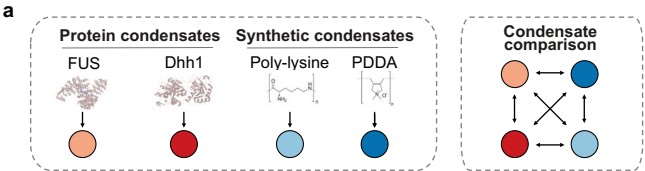

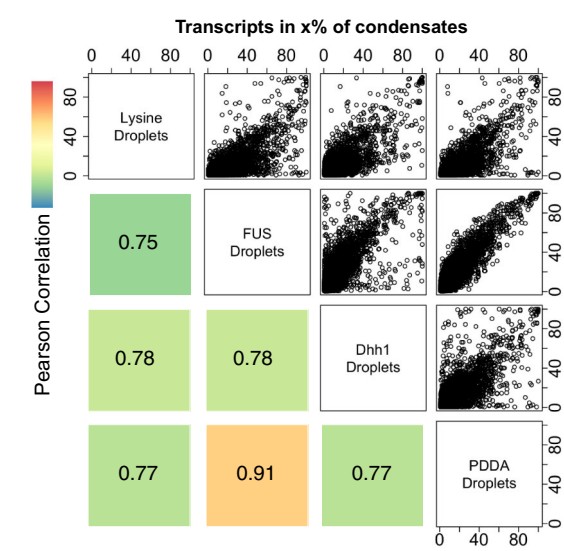

**Fig. 4 Comparison of RNA content across different coacervate and condensate types. a** Schematic representation of condensate types. Phase separation of synthetic condensates (CM-Dex:PDDA, CM-Dex:pLys) was induced through addition of carboxylmethyldextran (CM-Dex). **b** Scatter plots and corresponding Pearson correlations comparing how frequently each transcript is detected in different condensate types. Color represents magnitude of correlation.

Fig. 3a) since we observed that there are many enriched transcripts that are specific to the chemical composition of the condensate types (Supplementary Fig. 14b). This also enables us to mitigate batch effects due to differences in the input RNA. We saw that FUS and PDDA condensates cluster closely together, whereas lysine condensates clustered with Dhh1 droplets indicating close RNA content similarity between these condensate types (Supplementary Fig. 15a). The Dhh1 condensates as well as the CM-Dex:pLys coacervates split into two clusters which are distinguished by condensate size indicating that small and large Dhh1 and CM-Dex:pLys droplets enrich for different transcripts (Supplementary Fig. 15b, c and Supplementary Data 2). We also performed motif enrichment analysis for all condensate types and found that the most enriched motif of the PDDA condensates was also highly enriched in all other condensate types (Supplementary Fig. 14a, c). Hence, this motif might confer advantages for transcripts to be taken up into condensates universally, irrespective of the molecular composition of the condensate.

## Discussion

In summary, our data demonstrate for the first time that it is possible to explore the RNA content within single coacervate droplets. We dissected the molecular heterogeneity of a pool of coacervates allowing us to determine molecular differences between them. Thus far, differences between single coacervates could only be described on the phenotypic level by microscopy. Our ability to combine the sequencing data describing the RNA content with the FACS data describing the size and granularity of coacervates enabled us to link genotype and phenotype on the level of individual coacervates. Understanding the genotype-

phenotype link is of primary importance toward the generation of artificial cells, the origin-of-life and for modern biology[25].

A central question regarding the genotype of coacervates is, what types of RNAs it enriches for and which features the RNA molecules are characterized by. We found that RNAs with high sequence complementarity within or across RNA sequences are enriched in coacervates. This finding is reminiscent of the fact that stress granules, which form by liquid-liquid phase separation in cells, enrich non-coding RNAs which are complementary to mRNAs and likely form double-stranded RNA[13]. Hence, increased charge density as a consequence of RNA double strand formation might be a prevalent feature of RNA content in biomolecular condensates which can be recapitulated in in vitro reconstituted synthetic coacervate systems.

We further found that coacervates enrich for RNAs that contain sequence motifs that strongly resemble SINEs and in particular Alu elements. Interestingly, Alu element-containing RNAs were previously shown to be enriched in the nucleolus, the largest condensate in the cell nucleus of eukaryotic cells[26,27]. Our data therefore indicate that interactions of complementary Alu elements within transcripts could lead to formation of double-stranded RNA. This interaction, rather than overall differences in global RNA structure (Supplementary Fig. 6c) likely represents a key RNA feature that leads to enriched RNA localization into coacervates. Interestingly, other studies have also indicated that specific RNA sequences influence its role in phase-separated compartmentalization. For example, the RNA-mediated phase separation behavior for the SARS-Cov2 N-protein is strongly sequence specific for sequences at the 5′ end and sequences at the 3′ end which encodes the nucleocapsid RNA N-protein of the virus[28,29]. Specific sequences can also increase the demixing of RNA itself. This is exemplified by the phase separation of RNAs which contain expansions of the G4C2 repeats[30]. This hexanucleotide expansion is commonly associated with familial amyotrophic lateral sclerosis and frontotemporal dementia[31,32]. This is particularly interesting in the light of fact that this G4C2 repeat was shown to form G-quadruplexes in vivo and in vitro. This supports the view that sequence mediated RNA structure might be a strong determinant of RNA phase separation[33]. However, it should be noted that additional RNA independent factors which we did not investigate such as ionic strength[34] or peptide composition[35] might also influence RNA sequestration into coacervates. One example is a recent study that describes that ionic environment strongly influences the propensity for heterotypic peptide-RNA and homotypic RNA condensation[34]. These changes in the biophysical environment of the coacervates might ultimately not only influence RNA partitioning into phase-separated droplets but also alter the efficiency of catalysis as recently demonstrated for ribozymes[36].

When we compare the RNA content of protein-based condensate and synthetic polymer-based coacervates we found many similarities. Many transcripts that frequently enter one type of condensate also do so for others. Additionally, enriched transcripts for all condensate types are enriched for SINE sequence motifs, suggesting that these motifs might confer an advantage to condensate localization irrespective of the molecular composition of the condensate type. This is further supported by the fact that we also find these motifs in stress granule and p-body transcriptomes (Supplementary Fig. 12).

These results raise the question to which extent our results are biologically relevant beyond artificial coacervates systems. Several studies addressed the question if coacervates represent a suitable model for membraneless organelles since even simple RNA/polyamine coacervates recapitulate many features of condensates based on intrinsically disordered proteins[16]. In fact, there are several features which suggest that simple coacervate systems are an interesting model for biological condensates. For example, it has been shown that gene transcription is possible in CM-Dex:-pLys coacervates[21]. This is particularly interesting given the large number of recent findings deciphering the regulatory role of liquid-liquid phase separation in cellular transcription[37–40]. Furthermore, specific coacervate systems enable the formation of multiphase droplets with striking similarity to subcellular structures such as e.g. nuclear speckles[41,42]. Although it is obvious that simple synthetic coacervate system can never fully recapitulate the molecular complexity of membraneless organelles, our results present encouraging additional evidence that coacervates can approximate membranless organelles also on the transcriptomic level. Furthermore, it is conceivable to apply our single condensate sorting and sequencing methodology in the future to condensates isolated from cells as previously described[15] since only the FACS limits the size of condensates which can be sorted.

Taken together, our data demonstrate that single-cell RNA sequencing technology is not confined to the analysis of living cells but also applicable to RNA characterization of in vitro phase-separated coacervates[43]. It allows for highly multiplexed analysis of multiple condensate types and has the potential to uncover many aspects of the role of RNA in condensate formation with implications on several scientific disciplines from chemistry to cell biology. We envision that the approach will greatly facilitate the investigation of the complex roles of RNAs in phase separation as it enables the analysis of the transcriptomic complexity across a diverse pool of condensates and coacervates.

## Methods

**Single-coacervate sequencing protocol.** A detailed step-by-step protocol for single-coacervate sequencing can be found at https://doi.org/10.17504/protocols.io.bux5nxq6.

**Condensate generation.** Synthetic polymer-based coacervates were prepared as previously described (32). Specifically, Poly(Diallyl Dimethyl Ammonium Chloride) (PDDA, 8.5 kDa, monomer: 161.8 g mol$^{-1}$, Polyscience Inc.) or poly-L-lysine (4–15 kDa, monomer: 161.67 g mol$^{-1}$, Sigma) were mixed with CM-Dex sodium salt (10–20 kDa, monomer: 162.14 g mol$^{-1}$, Sigma) at a molar ratio of 6:1 (60 mM PDDA/pLys: 10 mM CM-Dex final concentrations) in Tris-MgCl$_2$ buffer (10 mM Tris-HCl pH 8.0 and 4 mM MgCl$_2$). Total RNA was isolated from iPSC cells (409B2) or MEF using the RNeasy Mini Kit (Qiagen). Coacervates were generated by adding CM-Dex, RNA and PDDA in the respective order to the Tris-MgCl$_2$ buffer to achieve a final RNA concentration of 50 ng/μl. For experiments using synthetic RNA, ERCC spike-in mix (Thermo Fisher) was spiked in to total RNA at a dilution of 1:80,000. Coacervates were then incubated for 1 h at room temperature while rotating before FACS sorting. FUS-GFP and Dhh1-mCherry proteins were cloned, purified and respective protein-based condensates were prepared as previously described (Supplementary Data 4) (4, 30). Recombinant FUS-GFP in 25 mM Tris-HCl pH 7.4, 150 mM KCl, 2.5% Glycerol and 0.5 mM DTT was used at a final protein concentration of 1 mg ml$^{-1}$. Dhh1-mCherry in 200 mM NaCl, 25 mM Tris (pH 7.4) and 10% glycerol was used at a final protein concentration of 150 μM. Dhh1 droplets were generated by adding ATP (final conc. 10 mM), creatin kinase based ATP recombination system CKM (40 mM ATP, 40 mM MgCl$_2$, 200 mM creatine phosphate, 70 U/ml Creatine Kinase), BSA (final conc. 1 mg ml$^{-1}$) and Hepes buffer (final conc. 50 mM) to the recombinant Dhh1-mCherry protein in low salt buffer (50 mM KCl, 30 mM HEPES-KOH pH 7.4, 2 mM MgCl$_2$).

**Single-coacervate index sorting.** RNA-containing coacervates (initial volume min. 250 μl) were sorted with a BD FACSAria Fusion flow cytometer using a 150 μl nozzle. Single-coacervates were index-sorted in precooled skirted twin.tec 96-well LoBind Plates (Eppendorf) containing 4 μl of 6 M Guanidine HCl (GuaHCl, Sigma) as lysis buffer. For each plate, one well was sorted with 1000 coacervates and one well was left empty as positive and negative controls respectively. Directly after sorting the plates were briefly spun down (max speed) to collect all FACS-derived droplets in the lysis buffer. The plates were then immediately put on dry ice until all other plates were sorted. Plates were kept at −80 °C until cDNA was prepared.

**Bulk coacervate FACS analysis.** Coacervates with and without RNA were analyzed on a BD FACSAria Fusion (150 mm nozzle) and data was processed in R using the *flowcore* package. For quantification of RNA incorporation into coacervates, size-matched FAM-labeled RNAs were synthesized (IDT) and incorporated into PDDA-CM-Dex coacervates as described above. Sequences of chemically synthesized oligos can be found in Supplementary Data 3.

**Confocal microscopy**. Image acquisition for experiments involving fluorescent coacervates was performed using a Olympus FV1200 confocal microscope using a ×20 oil immersion objective. For the estimation of the partition coefficients the ratio between [RNA concentration in droplets]/[RNA concentration in supernatant] was determined by quantification of fluorescence intensity of FAM (Supplementary Fig. 9d) or propidium iodide (Life Technologies) which was used to stain RNA (final concentration: 1 μg/ml) (Supplementary Fig. 1a).

**Single-coacervate library preparation**. Before library preparation the plates were spun down to collect all liquid at the bottom of the wells. SPRI beads (Agencourt RNAclean XP, Beckman Coulter) were equilibrated to room temperature and 2.2× SPRI beads were added to each well. Upon incubation for 5 min at room temperature, beads were washed twice using 80% EtOH as described in the SPRI bead manufacturer's protocol. EtOH traces were completely removed and beads were dried for 2–3 min (Note: Beads dry out fast after exposure to GuaHCl. Overdrying of beads will lead to significantly lower yields). RNA was eluted by resuspending beads in 3 μl of dNTP/oligodT mix, then beads were magnetically separated from RNA/dNTP/oligodT mix and transferred to a new 96-well plate. For library preparation of total RNA (Supplementary Fig. 2) oligodT priming was replaced by random priming. Next, the SMART-seq2 protocol described in Picelli et al.[11] was followed from step 9 onwards with the following modifications: (1) the template switching oligo was biotinylated on the 5′-end, (2) PCR preamplification was performed for 23 cycles. Size distribution of cDNA obtained from single coacervates was checked for randomly chosen samples to verify success of cDNA preparation. Next, tagmented libraries were prepared and sequenced (100 bp paired-end reads) on an Illumina HiSeq 2500 as described[11]. Libraries from single coacervates were also compared to input RNA (Fig. 3a). Library preparation of input RNA was performed as described in the original Smart-seq2 protocol using either 5 ng or 50 pg (Supplementary Fig. 6) of input RNA which were preamplified for 20 cycles.

**Data processing, quality control and analysis**. Raw sequencing data was processed using custom scripts and aligned to reference human transcriptome (hg38 sourced from Ensembl) using Kallisto (v0.44.0) with standard parameters including -pseudobam flag to obtain read coverage across each transcript. For data obtained from MEFs (Supplementary Fig. 11), raw sequencing data was aligned to the mouse genome (mm10) using STAR (v2.7.9a)[44]. Transcripts TPM values <1 were filtered out. For datasets with low average pseudoalignment (<40%), transcripts with less than 20% read coverage were excluded. Furthermore, since the coacervate size correlates with the number of transcripts detected as a consequence of coacervate size-dependent RNA concentrations we filtered out coacervates with <5% pseudoalignment for sizes FSC > 2e4. Enriched transcripts (Fig. 3a—red dots) were defined as transcripts whose residuals value was >30 when the data was fitted to a generalized additive model.

**Motif enrichment analysis**. De novo motif discovery was determined using MEME (v5.0.5) with the following parameters: -dna -time 18000 -mod anr -nmotifs 10 -minw 6 -maxw 50 -objfun classic -markov_order 2. Sequences obtained from the reference human transcriptome (hg38 sourced from Ensembl) were chosen as input for MEME analysis. The background was calculated using the sequences of all input transcripts. Enrichment of discovered motifs for each transcript was calculated using MAST (v5.0.5) with -nostatus -minseqs 21978 -remcorr -sep -ev 0.05 -c 1 parameters. MEME and MAST outputs were parsed for analysis in R using custom python scripts. Distances between every detected motif and its closest 5′-neighbor were calculated for each enriched transcript (from motif-start to motif-start—Supplementary Fig. 10). The same analysis was done for motifs detected in randomly non-enriched transcripts (Supplementary Fig. 10). Data was plotted using the ggplot2 and circlize packages.

**RNA folding analysis**. Analysis of minimum free energy for each enriched transcript and the same number of randomly selected non-enriched transcripts was performed using RNAfold (v2.4.12) with standard settings. RNAfold output was parsed for analysis in R using a custom python script.

**Comparison of sequence complementarity**. The presence of Motif 1 and its reverse complement Motif 2 on the same transcript (cis complementarity—Fig. 3d) was determined using MAST with E-values <0.05 as a cutoff. The complementarity of sequences across different transcripts (trans complementarity—Fig. 3d) was obtained by determining the pairwise local alignment using the Smith–Waterman algorithm. Briefly, two pools of transcripts were used for this analysis: enriched transcripts (residuals >30—see Fig. 3a) and the same number of randomly chosen non-enriched transcripts (residuals <30) with a similar transcript length distribution. Local-pairwise alignments for each transcript pair of the respective pools were calculated using the Biostrings package in R with the following parameters: $nucleotideSubstitutionMatrix(match = 2, mismatch = -1, baseOnly = TRUE)$, $pairwiseAlignment(gapOpening = -30, gapExtension = -0.05, scoreOnly = TRUE, type = "local")$. For comparison of enriched sequence motifs with SINEs (Fig. 3e) we obtained SINE reference sequences from RepBase (latest update: 08-24-2020). For each of the 10 consensus motifs, the 5 most significant motif hits found among

the enriched transcripts were compared to each SINE sequence by pairwise alignment using the Biostrings R package. Then the pairwise alignment score was averaged over the 5 most significant motifs for each consensus motif providing the alignment score displayed in the plot. Alignment parameters: $nucleotideSubstitutionMatrix(match = 2, mismatch = -1, baseOnly = TRUE); pairwiseAlignment(gapOpening = -10, gapExtension = 0, scoreOnly = TRUE)$.

**Analysis of stress granule and P-body transcriptomes**. Bulk transcriptome data from either stress granules (isolated from human U2OS cells) or P-bodies (isolated from human HEK293 cells) was analyzed for motif enrichment using MEME (v5.0.5). Prior to the analysis, genes with FPKM/CPM values <1 were filtered out. For stress granule data, gene names were converted to transcript IDs by using the most dominant transcript for each enriched gene in human tissues[45]. The 100 most enriched transcripts in stress granules and p-bodies were then used for motif enrichment analysis as described above.

**Dimensionality reduction and differential expression analysis**. Analysis of transcript-based coacervate heterogeneity (Supplementary Fig. 15a, b) was conducted using the Seurat R package (v3.1.5). For this analysis, we used transcripts as input which were enriched for each coacervate type as defined by the analysis in Fig. 3a (residuals >30 when fitted to a generalized additive model). These transcripts were normalized to the TPM values of the transcripts in the input RNA pool and subsequently scaled within each experiment. For clustering and UMAP analysis the first five principal components were used.

For differential gene expression (DE) analysis (Supplementary Fig. 15c) we focused on the differences between the two main clusters (Lysine/Dhh1 vs. FUS/PDDA). In order to find DE genes, we used the FindMarkers function of the Seurat R package with a Bonferroni-adjusted p value significance threshold of <0.01.

**Reporting summary**. Further information on research design is available in the Nature Research Reporting Summary linked to this article.

## Data availability

The data supporting the findings of this study are available from the corresponding authors upon reasonable request. The sequencing data generated in this study have been deposited in the ArrayExpress database under accession code E-MTAB-11348.

## Code availability

The computational code used in this study is available at GitHub (https://github.com/wollnylab/single_coacervate_seq) or upon request.

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

## Acknowledgements

We would like to thank Malgorzata Santel, Anne Weigert and Theresa Schaffer for FACS sorting condensates. Further, we thank Barbara Schellbach and Antje Weihmann for performing Illumina sequencing. We also want to thank Hannes-Claudius Schulze for experimental help, Wulf Hevers for help with microscopy experiments and Tobias Gerber, Sabina Kanton, Agnieska Brazovskaja, Stephan Bernhart, Jörg Fallmann and the Treutlein/Camp labs for helpful discussions. This work was supported by core funds of the Max Planck Society and ETH Zurich (B.T.), by ERC StG 758877 ORGANOMICS (B.T.), the Volkswagen Foundation (T.-Y.D.T.) as well as by ERA-NET rare disease research implementing IRDiRC objectives—No. 643578—Grant: REPETOMICS (B.T. and D.W.)

## Author contributions

D.W. performed the experiments with help from F.A. and J.M.; D.W., B.V., A.S. and Z.H. performed data analysis; J.W. and A.H. provided recombinant FUS-GFP; M.H. and K.W. provided recombinant Dhh1-mCherry; D.W., J.G.C., T.-Y.D.T. and B.T. designed experiments and wrote the manuscript.

## Competing interests

The authors declare no competing interests.
