## [Peer Review File · Nature Communications]

Title: Characterization of RNA content in individual phase-separated coacervate microdropletsREVIEWER COMMENTS

Reviewer #1 (Remarks to the Author):

This manuscript develops methods to sequence the composition of individual phase-separated droplets containing RNA. The impact of the work is to present a possible workflow to show how individual assemblies could be sequenced. However, the work is limited in three manners (detailed below): a) insufficient details are provided to understand how experiments are performed and validated, b) the work is not explained clearly, making it a burden on the reader to slog through what was done and how it is interpreted, and c) since the work is all done on artificial systems, there is limited impact on our understanding of biology. My recommendation is that the work is re-written to include relevant information and with improved clarity, and then the work could be considered as a technical achievement that might be appropriate for publication in Nature Communications.

This review is from Roy Parker and I would be willing to discuss these issues with the authors if they desire.

Specific Issues:

1) The entire approach rests on the success of sorting droplets into 96 well plates.

Issue a) I assume a mixture is created that will form RNA containing coacervates, and then that mixture is run through the sorter. However, there will be a liquid phase and a droplet, and when a droplet is sorted, I assume some volume of the liquid phase will also be sorted with it. Thus, the sequencing is actually a mix of droplet and liquid phase. Is this true? If not, how are the coacervates sorted to avoid carrying over any bulk liquid phase?

b) It's unclear what metrics are used by the FACS sorter to sort the condensates. Is it a combination of forward and side scatter? I recommend the authors provide the flow cytometry data. Are these metrics then used to show that no condensates can be sorted in a negative control sample (no condensates) so one knows they are sorting bona fide condensates?

c) In order to understand the significance of the RNA composition, more information should be provided on the nature of the coacervates used should be provided. Specifically, i) Are these coacervates forming independent of RNA, and then RNA is recruited to them, or is RNA required for their formation? (Minor note: I tried to find the reference for the polymer-based assemblies and it was not provided (noted as Ref 32 in the methods). ii) What is the partition coefficient of RNA into these assemblies? This seems useful in thinking about how the assembly is forming and what different sequence elements mean. iii) At least once, it would be useful to look at the total composition of the RNA in the assemblies. I suggest this since rRNA is ~90% of the total RNA being used and to only sequence poly(A)⁺ RNA might give a misleading view of the RNAs recruited into these assemblies. iv) It would be useful to point out how these assemblies relate to biological condensates, and to point out the limitations of these simplified systems.

2) Another key issue is how the relative abundance of RNA in the coacervates is determined relative to

the bulk. Is this based on similar sequencing of the total RNA? I assume so, but details like this are missing throughout the manuscript, which makes each experiment difficult to interpret. I suggest the authors go over the manuscript and methods and make sure that the methodology and logic for each experiment are clearly explained and provided in the work.

3) One of the major conclusions of this paper are particular motifs are enriched for all condensate types. Quote, “this motif might confer advantages for transcripts to be taken up into condensates universally, irrespective of the molecular composition of the condensate.” If this is true, one prediction would be that this motif should be enriched in the transcriptome of p-bodies and/or stress granules (Khong et al., 2017; Hubstenberger et al., 2017; Matheny et al., 2020; Namkoong et al., 2018). I think the authors should do this analysis in order to connect the in vitro work with experiments done in cells.

4) The experiments testing the enrichment of specific motif containing oligomers are interesting. However, it would be important to a) provide some sense of what the normalized FAM signal means: what is it normalized to?, would other RNAs show for enrichment? what kind of partition coefficient does this correspond to (which would be assessed easily by microscopy)? and b) Are these differences statistically significant?

Minor Comment:

5) I find it very surprising that length does not have any impact on recruitment into these coacervates for two reasons. First, in all the analysis of RNP granules from cells, or artificially created RNA self-assemblies, RNA length is a significant metric for inclusion due to its increase in valency with longer length (e.g. Van Treeck et al., 2018, PNAS). Second, analysis of fundamental properties of coacervates would also suggest that length should matter as long as there are any interactions that increase partitioning of the macromolecule into the coacervate (Banini et al., 2016, Cell).

I suggest two additions:

a) Can the authors validate that the RNA is full length when it enters the assembly reaction? (Perhaps show a tape station trace of the bulk RNA before and after coacervate formation). This would remove any concerns about a loss of a length effect due to the shortening of bulk RNA.

b) Provide some type of explanation why length would not be a factor in recruitment under these types of coacervates.

Reviewer #2 (Remarks to the Author):

The authors have developed single-coacervate RNA-sequencing technique to investigate the characteristic of RNAs preferentially associated with coacervates, finding that SINE sequences with double-strand structure are frequently incorporated into the coacervates. The single-coacervate sequencing technique presented from the study could provide novel way to assess the characteristic of coacervate formation in detail. However, there are some important questions to be addressed for the

main conclusions from the study:

(1) Control for subsampling effect of total RNA

For the preferential association analysis shown in Fig. 3, the authors have used input RNA abundance as a reference. However, sequencing library generated from low quantity RNA can have unique bias. Thus the most suitable background would be the library generated from the diluted input RNA (concentration and amount similar to the level in single coacervates). Could authors explain whether this kind of bias has been taken into account or perform suitable control experiment with serial dilution of input RNA? Data from the paper (<https://www.ncbi.nlm.nih.gov/pmc/articles/PMC3467340/>) could be used.

(2) Usage of random/designed RNA sequence

Using human total RNA as a starting material could pose an intrinsic bias for the study as the sequence composition of human total RNA is heavily skewed. To understand the intrinsic property of RNA being incorporated into coacervate, series of designed RNAs to represent different situations (different length, AU contents, reverse complements and hairpin structures) can be synthesized or in vitro transcribed from the designed DNA material. Or the authors could use total RNA from other species, and show that the conclusion (enrichment of dsRNA structure) can be validated in such situation.

(3) Differential analysis between coacervates,

Fig S9 demonstrates the power of the method, which is to differentiate coacervates formed from different materials. Could authors elaborate more on this analysis by showing what kind of RNAs are generating such a difference between condensate types?

Reviewer #3 (Remarks to the Author):

In this manuscript, Wollny et al. utilize high-throughput sequencing of mRNA for protein-based condensates (FUS and Dhh1) as well as for synthetic ones (coacervates from PDDA/PLys with Cm-Dextran). The employment of fluorescence activated cell sorting allows them to quantify RNA content up to a single coacervate. They analyze the dependence of genomic mRNA (transcript) uptake on parameters such as condensate droplet size, number and type. The authors show a profound statistical analysis of the transcripts partitioned inside, hereby examining the RNA length, its abundance and elaborate structural motifs that enhance or decrease uptake into the condensate phase. They conclude that

- 1) Higher amount of RNA prior to droplet formation leads to larger uptake into the droplets
- 2) Enrichment of transcripts into the droplets when the RNA contains complementary sequences and motifs that resemble short interspersed elements
- 3) These enriched motifs are present in both protein-based as well as synthetic condensates, demonstrating that uptake of specific motifs might not be influenced by other factors that distinguish the studied droplet types and also highlighting that coacervates can be a great model to understand condensate behavior.

I do think that this manuscript contains valuable insights for the community. The fact that synthetic condensates show many similarities in partitioning RNA, e.g. what type of sequences are enriched, makes it a great control in condensate research. I also think it may have great implications on the origin of life, hereby strengthening the role of coacervates as a relevant protocell model.

I do, however, have several concerns:

1. Many of the described findings seem to be less surprising but more of a logical nature based on previous work. The fact that smaller droplets contain rather longer transcripts (p.4) seems intuitive, and as the droplets get bigger, they can harbor more RNA resulting in an increase in diversity in bigger droplets. Increased uptake of RNA due to larger abundance in the RNA pool is what you would expect, too.

Generally, one would expect that dsRNA should have a higher charge density than ssRNA and therefore a bigger propensity to partition into the droplets. Whether the RNA really remains double-stranded inside the droplet would necessitate further studies, e.g. FRET (Cakmak et al 2020) as the coacervate/condensate interior might destabilize (DNA-)hybridization. (Nott et al 2016, Vieregge et al 2018)

Coacervates as models for membraneless organelles / biomolecular condensates have been proposed by many groups, eg. Keating, Spruijt,..., I agree that this manuscript strengthens the role of coacervates as condensate models, but I would disagree with statements like "This provides for the first time a direct link between synthetic coacervates and biomolecular condensates in cells, implying that coacervates can serve as models of biological systems." Particularly because there are also groups like the one from Priya Banerjee who compared protein-based condensates with coacervates in their work.

2. 22 references is not a lot, considering that the major impact of this work comes from the experimental setup that should benefit fields such as origin-of-life or modern biology. I wonder whether the authors could better put their results in perspective to what is already known (see above) and highlight better which findings actually were not expected based on previous work. I also think the authors could better outline in the conclusion section, how their experimental setup might solve issues that other groups had so far (eg. By referring to difficulties of in vivo/vitro analysis of condensates, or maybe in respect to synthetic coacervates in the origin of life field..)

3. There are many other parameters (apart from the mentioned potential hybridization) that affect uptake into the droplets. Eg. charge density, ionic strength, amino acid sequence of the main droplet building blocks, viscosity/surface tension,... Now you could argue that, even though neglected here, specific transcripts/motifs were commonly enriched, indicating that these other parameters might be not as important for the uptake of some transcripts. But given the fact that condensates and coacervates might have significant differences in the described parameters (eg. Recent review from Spruijt), a more detailed consideration would help this work.

Some minor things:

On p4: "In comparison coacervates containing the longest average transcript length were among the

smallest coacervates.”

On p5: “Our analysis showed that there was no correlation to the transcript length and its frequency in detection into the coacervates”

Not sure if I got it right, but seems contradicting.

In the SI, the materials part is missing.

Generally, some plots (both text and SI) are missing units on the axis

S1d is missing, S5b seems to be missing something based on the caption?

In the SI, when you explain how you prepare the droplets, it would be great if one could better compare the ionic strength and charge density between condensates and coacervates, eg. either express the polycations/anions as mg/mL or maybe better mM of charge for all the types.

SUMMARY

We would like to thank all 3 reviewers for the thorough and constructive feedback on our manuscript. We strongly believe that the comments and suggestions greatly improved the quality of our manuscript. Here, we summarize the most important additions / changes which advanced the manuscript we originally submitted:

- We performed single coacervate sequencing using RNA from a different species (mouse) as input to which revealed enrichment of dsRNA structure driven by mononucleotide repeats rather than SINE elements
- We repeated single coacervate sequencing using defined quantities of synthetic RNA (ERCC) to validate our results and quantify the detection limit
- We analyzed published datasets from stress granules and p-bodies and compared these findings to our data which demonstrated *in vivo* relevance of our findings
- We sequenced total RNA from coacervates (in addition of poly-A transcripts) to provide a more comprehensive overview of the RNA content of single coacervates
- We reexamined the finding that length is not a strong predictor for the recruitment of RNA into coacervates and provide additional data and novel analyses to clarify the nature of our findings
- We performed microscopy experiments to validate our findings and to provide partition coefficients
- We added differential gene expression analysis to reveal which genes are preferentially assembling in different coacervate / condensate types
- We examined the sensitivity of our results to input RNA concentration by sequencing input samples at different concentrations
- We have corrected numerous minor mistakes, provided additional information to clarify the text and extended our discussion to help the reader to put our work in the context of the previous findings

Below, we detail the findings as a result of the feedback

REVIEWER COMMENTS

Reviewer #1 (Remarks to the Author):

This manuscript develops methods to sequence the composition of individual phase-separated droplets containing RNA. The impact of the work is to present a possible workflow to show how individual assemblies could be sequenced. However, the work is limited in three manners (detailed below): a) insufficient details are provided to understand how experiments are performed and validated, b) the work is not explained clearly, making it a burden on the reader to slog through what was done and how it is interpreted, and c) since the work is all done on artificial systems, there is limited impact on our understanding of biology. My recommendation is that the work is re-written to include relevant information and with improved clarity, and then the work could be considered as a technical achievement that might be appropriate for publication in Nature Communications.

This review is from Roy Parker and I would be willing to discuss these issues with the authors if they desire.

First of all, we would like to thank Dr. Parker for acknowledging our technical achievement and its potential for the future of the field. We have taken considerable effort (detailed below) in increasing the understandability of the text and hope that the new version is significantly improved in that regard.

Specific Issues:

1) The entire approach rests on the success of sorting droplets into 96 well plates.

Issue a) I assume a mixture is created that will form RNA containing coacervates, and then that mixture is run through the sorter. However, there will be a liquid phase and a droplet, and when a droplet is sorted, I assume some volume of the liquid phase will also be sorted with it. Thus, the sequencing is actually a mix of droplet and liquid phase. Is this true? If not, how are the coacervates sorted to avoid carrying over any bulk liquid phase?

It is correct that the droplet that is created by the FACS machine will always contain a mixture of the coacervate and its surrounding liquid. This is technically impossible to avoid. We added this information to the manuscript now to avoid confusion for the reader (see page 4).

However, our data indicates that the surrounding liquid is not a major contributor to the sequencing results obtained after FACS sorting. In Fig. 1c one can see that larger coacervates (large circles) contain a higher

number of transcripts (because larger coacervates will contain more RNA). If the majority of information would come from the surrounding liquid, one would not expect to see such a correlation. In order to illustrate this more clearly, we added another plot (see new Fig. S1g) which demonstrates the correlation between coacervate size and the number of detected genes more directly.

b) It's unclear what metrics are used by the FACS sorter to sort the condensates. Is it a combination of forward and side scatter? I recommend the authors provide the flow cytometry data. Are these metrics then used to show that no condensates can be sorted in a negative control sample (no condensates) so one knows they are sorting bona fide condensates?

We indeed used a combination of forward and side scatter to FACS gating. The only additional gate is a doublet exclusion gate. We now include the flow cytometry data (see new Fig. S1b) and clarified this in the text (see page 3).

Further, we previously already implemented the suggested no condensate negative control for all plates sorted in this study to ensure that the libraries are derived from coacervates (see bioanalyzer traces in Fig. S1d). In order to strengthen this point, we now included comparisons of cDNA yield upon library preparation from single coacervates compared to the no coacervate negative control (see new Fig. S1f). This analysis quantitatively demonstrates markedly more cDNA in the well in which single coacervates were sorted. The low amount of cDNA (~70pM) in the no coacervate negative control stems from the leftover primers used for library preparation (see Fig. S1d – primer peak at ~100 bp)

c) In order to understand the significance of the RNA composition, more information should be provided on the nature of the coacervates used should be provided.

- Specifically, i) Are these coacervates forming independent of RNA, and then RNA is recruited to them, or is RNA required for their formation? (Minor note: I tried to find the reference for the polymer-based assemblies and it was not provided (noted as Ref 32 in the methods).

We wish to apologize for providing wrong reference number. The references have now been updated. The reason why we chose this coacervate system is indeed because it has been carefully characterized by one of the authors (T.Y.D. Tang – see Tang et al., Nat Chem 2014 / Beneyton et al., ChemSysChem 2020). We additionally added more information in the text describing that the coacervates form with or without RNA (see text on page 3).

- ii) What is the partition coefficient of RNA into these assemblies? This seems useful in thinking about how the assembly is forming and what different sequence elements mean.

We have now measured the partition coefficient of RNA into PDDA/CM-Dextran coacervates by fluorescence microscopy (mean part. coeff. = 9.46, SD = 2.08) and included the results in the new Fig.S1a.

- iii) At least once, it would be useful to look at the total composition of the RNA in the assemblies. I suggest this since rRNA is ~90% of the total RNA being used and to only sequence poly(A)+ RNA might give a misleading view of the RNAs recruited into these assemblies.

This is a very good point. In order to address this issue, we sequenced RNA from 1000 sorted coacervates and prepared libraries using a random hexamer instead of an oligo-dT primers. This change in the library preparation protocol ensures reverse transcription of all RNAs in the coacervates in comparison to the polyadenylated transcripts (which are mostly mRNAs) that we profiled in all other experiments. We find that the overall composition of RNA biotypes in droplets in comparison to the Input RNA is quite similar to the previous results (see new Fig. S2 as well as new Table S1). Dr. Parker was right in the assumption that by sequencing only polyadenylated transcripts we have missed mostly rRNA. However, as we pointed out in the main text (see page 4) we deliberately focused on mRNAs, since they represent a more complex pool of transcripts in terms of sequence and length diversity.

- iv) It would be useful to point out how these assemblies relate to biological condensates, and to point out the limitations of these simplified systems.

Although some work regarding this question has been done by others, how the specific coacervate system that we used in this study relate to biological condensates was still an open question which in part motivated our study. We believe that our data in Fig. 4 as well as the new data of p-bodies and stress granule comparisons to our system (see point 3 below) provide novel insights into this question. Yet, we agree that our coacervate system is very much a simplified system compared to cellular condensates which obviously represents a more molecular complex system. We have therefore additionally expanded our discussion (see page 11) regarding relationship between our system and biological condensates.

2) Another key issue is how the relative abundance of RNA in the coacervates is determined relative to the bulk. Is this based on similar sequencing of the total RNA? I assume so, but details like this are missing throughout the manuscript, which makes each experiment difficult to interpret. I suggest the authors go over the manuscript and

methods and make sure that the methodology and logic for each experiment are clearly explained and provided in the work.

We again apologize for not being clear enough regarding this point. The term “relative abundance” actually refers to the TPM values obtained after sequencing. We used the term relative abundance in the hope of circumventing technical (bioinformatics) jargon for an audience that might be unfamiliar with it. However, it appears as if this only led to more confusion.

TPM stands for “transcripts per million”. This way of quantifying transcripts abundance normalizes for transcript length and sequencing depth (like FPKM/RPKM) but has the additional feature that the sum of normalized reads in each sample/coacervate is equal. Simply speaking, the TPM value can be read as: for every million of transcript molecules in the library, there are x copies of a given transcript (very much alike percent – which is a fraction of 100 instead of 1 million). Hence, the term relative abundance actually refers to “relative to all other transcripts in the library / coacervate”.

We decided to eliminate the term “relative abundance” from our manuscript. We instead mention “TPM” directly and (on top of our own explanation) refer to the paper originally describing TPM (Wagner et al., 2012) which very nicely and intuitively describes the TPM metric.

3) One of the major conclusions of this paper are particular motifs are enriched for all condensate types. Quote, “this motif might confer advantages for transcripts to be taken up into condensates universally, irrespective of the molecular composition of the condensate.” If this is true, one prediction would be that this motif should be enriched in the transcriptome of p-bodies and/or stress granules (Khong et al., 2017; Hubstenberger et al., 2017; Matheny et al., 2020; Namkoong et al., 2018). I think the authors should do this analysis in order to connect the in vitro work with experiments done in cells.

We would like to thank Dr. Parker for this interesting suggestion, as it provides a very straight forward way to test the in vivo relevance of our findings. We have taken data from stress granules (Khong et al., 2017 – U2OS cells) as well as p-bodies (Hubstenberger et al., 2017 – HEK293 cells) and looked for the most enriched motifs in both datasets (see new Fig. S12).

Interestingly, we found the most enriched motif from our PDDA/CM-Dextran coacervates also among the Top10 most enriched motifs in stress granules (#10). Further, we found its reverse complement (2nd most enriched motif in our data) also to be highly enriched in p-bodies (#2). We believe that these findings are striking for two reasons:

- a) Considering that stress granules & p-bodies are significantly more molecularly complex compared to our in vitro coacervate system, it was conceivable that factors other than the occurrence of our motifs would drive the assembly of RNA into condensates in vivo. Yet, the motifs we discovered appear to be also relevant for RNA assembly into condensates in cells.*
- b) Beyond the motifs that we discovered in the course of this study, the analysis suggested by Dr. Parker uncovered a number of novel motifs enriched in stress granules / p-bodies which (to the best of our knowledge) have not been described before. These motifs represent an interesting starting point for future studies exploring their role in different condensate types.*

4) The experiments testing the enrichment of specific motif containing oligomers are interesting. However, it would be important to a) provide some sense of what the normalized FAM signal means: what is it normalized to?, would other RNAs show for enrichment? what kind of partition coefficient does this correspond to (which would be assessed easily by microscopy)? and b) Are these differences statistically significant?

Regarding a)

We apologize for accidentally omitting a description of what the signal was normalized to. For this experiment, the FACS machine measured the fluorescence intensity of every coacervate irrespective of how large the coacervates are. Since large coacervates will always show a stronger intensity (because more RNA is present in them), we normalized the fluorescence intensity to the size of coacervate to account for this confounding factor. Further, we measured partition coefficients (mean part. coef. = 10.2, SD=1.01) via microscopy. While performing the experiment, we were additionally able to confirm our FACS data by quantifying the fluorescence intensity of images from coacervates containing either double-stranded RNA (motif1 AND motif2) or single stranded RNA (motif1 OR motif2) (see new Fig. S9d).

Regarding b)

All differences are highly statistically significant $p < 2e-16$ (Bonferroni adjusted p value)

Minor Comment:

5) I find it very surprising that length does not have any impact on recruitment into these coacervates for two reasons. First, in all the analysis of RNP granules form cells, or artificially created RNA self-assemblies, RNA length is a significant metric for inclusion due to its increase in valency with longer length (e.g. Van Treeck et al., 2018, PNAS). Second, analysis of fundamental properties of coacervates would also suggest that length should matter as long as there are any interactions that increase partitioning of the macromolecule into the coacervate (Banini et al., 2016, Cell).

I suggest two additions:

a) Can the authors validate that the RNA is full length when it enters the assembly reaction? (Perhaps show a tape station trace of the bulk RNA before and after coacervate formation). This would remove any concerns about a loss of a length effect due to the shortening of bulk RNA.

We have included examples of RNA length distribution before coacervate formation (Input RNA) as well as after coacervate formation (1000 coacervates) as assessed by bioanalyzer traces. When comparing these distributions, we do not observe any change in length before and after the experiment. (see new Fig. S7c)

b) Provide some type of explanation why length would not be a factor in recruitment under these types of coacervates.

Given the published data, we were indeed also surprised to no clear correlation between RNA length and how frequently these RNAs are found in coacervates (Pearson's correlation coefficient $r = 0.06$ - see updated Fig. S7b). Our interpretation of the data was that the effect of how abundant RNAs are in the input pool which we used for our experiment dominates over most other variables (such as length). Simply speaking: If an RNA molecule A is vastly more abundant than RNA molecule B, then molecule A will be a lot more frequently found in the coacervates irrespective differences in length. In fact, we also observed this effect when we used synthetic RNAs of varying length and sequence as requested by Reviewer #2 (see new Fig. S5).

We then hypothesized that if our interpretation is right, we should see an increase of the correlation between RNA length and how often the RNAs are detected in coacervates once we normalize for abundance. We therefore analyzed the correlation between RNA length and enrichment in coacervates only for RNA molecules that similarly abundant in the input pool (Input TPM bins). We indeed found that for most bins there is a higher (and almost exclusively a positive) correlation between RNA length and enrichment into coacervates compared to when we take the whole input into consideration (see new Fig. S7d).

Reviewer #2 (Remarks to the Author):

The authors have developed single-coacervate RNA-sequencing technique to investigate the characteristic of RNAs preferentially associated with coacervates, finding that SINE sequences with double-strand structure are frequently incorporated into the coacervates. The single-coacervate sequencing technique presented from the study could provide novel way to assess the characteristic of coacervate formation in detail. However, there are some important questions to be addressed for the main conclusions from the study:

We want to thank the reviewer for acknowledging the novelty of our approach.

(1) Control for subsampling effect of total RNA

For the preferential association analysis shown in Fig. 3, the authors have used input RNA abundance as a reference. However, sequencing library generated from low quantity RNA can have unique bias. Thus the most suitable background would be the library generated from the diluted input RNA (concentration and amount similar to the level in single coacervates). Could authors explain whether this kind of bias has been taken into account or perform suitable control experiment with serial dilution of input RNA? Data from the paper (<https://www.ncbi.nlm.nih.gov/pmc/articles/PMC3467340/>) could be used.

It was noticed correctly, that the RNA input we used was at a higher concentration (5ng) compared to the amount of RNA present in single coacervates. The reason why we decided for a higher concentration was that we believe it is important that the sequenced input represents the “ground truth” with regards to which RNAs were added to the droplets in the first place. For instance, if we sequenced a diluted input sample, it might be problematic to answer the question if certain RNAs are excluded from coacervates. In order to answer this question, we need to find transcripts that are present in the input but not in coacervates. By diluting the input before sequencing it, we might miss a number of transcripts leading to the false conclusion that some transcripts are not found in coacervates because they were not present in the input.

We do, however, understand that it might be important to understand how much the results vary if one wishes to decrease the input amount. Hence, we performed an experiment where we sequenced single coacervates and compared them to 5ng RNA input (as before) as well as to 50pg RNA input (100-fold dilution). When comparing the two input samples, we did see a high correlation between high (5ng) and low (50pg) input concentrations (pearson’s correlation coefficient $r = 0.86$; see new Fig. S6a). We then repeated the analysis from Fig. 3A which detected transcripts that enter coacervates of high frequency and are enriched in SINE motifs, since a lot of important conclusions of the paper are based on this analysis. We obtained highly similar results when taking 5ng input compared to 50pg input (see new Fig. S6b-d). Thus, our results are robust across at least two orders of magnitude of input abundance, which is important information from a technical point of view. Yet, we would like to stress that we still believe that sequencing the input at high concentration is preferable, as described above.

(2) Usage of random/designed RNA sequence

Using human total RNA as a starting material could pose an intrinsic bias for the study as the sequence composition of human total RNA is heavily skewed. To understand the intrinsic property of RNA being incorporated into coacervate, series of designed RNAs to represent different situations (different length, AU contents, reverse complements and hairpin structures) can be synthesized or in vitro transcribed from the designed DNA material. Or the authors could use total RNA from other species, and show that the conclusion (enrichment of dsRNA structure) can be validated in such situation.

We would like to thank Review#2 for this interesting suggestion. Our initial manuscript already contained data from synthesized RNA species as an independent confirmation of our results obtained by sequencing human total RNA (see old Fig. S6). Yet, in order to more directly address Reviewer#2’s concern, we decided to perform an experiment in which we added RNA from a different species (mouse embryonic fibroblasts (MEF)) which also included a synthetic RNA mix (ERCC spike in mix - a very established and frequently used for sequencing experiments) to our coacervates. We were particularly intrigued by the suggested experiment that used MEF-derived RNA because one of the main findings of our paper is that RNAs which are enriched in coacervates contain SINES, in particular ALU elements (See. Fig. 3e). ALU elements are, however, primate specific – so they should not appear in our motif enrichment analysis when we use MEF-derived RNA. When performing the experiment, we again observed that most RNAs enter coacervates at high frequency strongly depended on how abundant these RNAs were in the input (see new Fig. S11a). We also again observed outlier transcripts which enter coacervates at higher frequency than expected given their input abundance. When we performed motif enrichment analysis we indeed did not observe any sequences that resemble SINE motifs. Instead, we saw a striking enrichment of A and T stretches (see new Fig. S11b – Motifs 2,3,7) as well as G and C stretches (Motifs 1,4,8). This result again strongly suggests the enrichment of dsRNA structure of mouse-derived RNA in coacervates. Yet, in contrast to human RNA, the complementarity appears not to driven by SINE elements but rather by mononucleotide repeats.

When analyzing the synthetic RNA mix we observed that all RNAs are enriching in coacervates as a function of their input abundance irrespective of their length (see new Fig. S5). This result again confirms the finding that the input RNA amount is a very strong determinant of RNA assembly into coacervates (compare to

Fig. 3a). Additionally, this experiment allowed us to determine the detection sensitivity of our approach, as we could show that we reliably detect transcripts from a concentration of 1fM onwards. We did not observe outlier transcripts indicating little complementarity in this pool of sequences. We believe that these results significantly enrich our manuscript as they independently confirm several conclusions of our work.

(3) Differential analysis between coacervates,

Fig S9 demonstrates the power of the method, which is to differentiate coacervates formed from different materials. Could authors elaborate more on this analysis by showing what kind of RNAs are generating such a difference between condensate types?

We agree with Review #2 that our old Fig. S9 (now Fig. 15) would benefit from more information about which genes contribute to the differential clustering of different coacervate types. Our UMAP analysis of old Fig S9 points towards a clear separation between two groups: PDDA/FUS coacervates vs. Dhh1/Lysine coacervates. We performed differential genes expression analysis to find genes which significantly ($p < 0.01$ - bonferroni correction) differ between these groups. We found a number of genes which are explaining the differences between the groups which we display in the new Fig. S15c. In addition, we supplement a table (new Table S2) which details all information the Fig. S15c is based on.

Reviewer #3 (Remarks to the Author):

In this manuscript, Wollny et al. utilize high-throughput sequencing of mRNA for protein-based condensates (FUS and Dhh1) as well as for synthetic ones (coacervates from PDDA/PLys with Cm-Dextran). The employment of fluorescence activated cell sorting allows them to quantify RNA content up to a single coacervate. They analyze the dependence of genomic mRNA (transcript) uptake on parameters such as condensate droplet size, number and type. The authors show a profound statistical analysis of the transcripts partitioned inside, hereby examining the RNA length, its abundance and elaborate structural motifs that enhance or decrease uptake into the condensate phase. They conclude that

- 1) Higher amount of RNA prior to droplet formation leads to larger uptake into the droplets
- 2) Enrichment of transcripts into the droplets when the RNA contains complementary sequences and motifs that resemble short interspersed elements
- 3) These enriched motifs are present in both protein-based as well as synthetic condensates, demonstrating that uptake of specific motifs might not be influenced by other factors that distinguish the studied droplet types and also highlighting that coacervates can be a great model to understand condensate behavior.

I do think that this manuscript contains valuable insights for the community. The fact that synthetic condensates show many similarities in partitioning RNA, e.g. what type of sequences are enriched, makes it a great control in condensate research. I also think it may have great implications on the origin of life, hereby strengthening the role of coacervates as a relevant protocell model.

We would like to thank the reviewer for acknowledging the value our manuscript adds to the community.

I do, however, have several concerns:

1. Many of the described findings seem to be less surprising but more of a logical nature based on previous work. The fact that smaller droplets contain rather longer transcripts (p.4) seems intuitive, and as the droplets get bigger, they can harbor more RNA resulting in an increase in diversity in bigger droplets. Increased uptake of RNA due to larger abundance in the RNA pool is what you would expect, too. Generally, one would expect that dsRNA should have a higher charge density than ssRNA and therefore a bigger propensity to partition into the droplets. Whether the RNA really remains double-stranded inside the droplet would necessitate further studies, e.g. FRET (Cakmak et al 2020) as the coacervate/condensate interior might destabilize (DNA-)hybridization. (Nott et al 2016, Vieregg et al 2018)
Coacervates as models for membraneless organelles / biomolecular condensates have been proposed by many groups, eg. Keating, Spruijt,..., I agree that this manuscript strengthens the role of coacervates as condensate models, but I would disagree with statements like "This provides for the first time a direct link between synthetic coacervates and biomolecular condensates in cells, implying that coacervates can serve as models of biological systems." Particularly because there are also groups like the one from Priya Banerjee who compared protein-based condensates with coacervates in their work.

We agree with Reviewer #3 that many of our findings are of logical nature based on the expectations build on previously published work as well as intuition. However, we do believe that this is a positive sign. Given that we describe the application of single cell sequencing technology to phase-separated condensates and coacervates which has never been done before, one natural first questions that one might raise is: "Is it technically possible?". The fact that many of our findings agree with previous work was a very encouraging sign to us, as it helped to build trust in the results we obtained with our method.

We would also like to point out that, although some results like the relationship between coacervate size and the average length of RNAs (or the complexity of RNAs) within coacervates might potentially be considered intuitive, their relationship could not have been quantitatively tested without single coacervate/condensate resolution. Regarding the point that we make about coacervates as models for membraneless organelles / biomolecular condensates: We admit that our statement was an overstatement and subsequently changed it. We now write that our findings strengthen the role for coacervates as models for biomolecular condensates. We performed additional analysis which further support this point. Specifically, we analyzed data from RNA content of stress granules (Khong et al., 2017 – U2OS cells) as well as p-bodies (Hubstenberger et al., 2017 – HEK293 cells). We then looked if the most enriched motif from our PDDA/CM-Dextran coacervates could also be detected among the RNAs found in stress granules / p-bodies in vivo. Interestingly, we found this motif to be among the Top10 most enriched motifs in stress granules (#10) and its reverse complement (2nd most enriched motif in our data) to be highly the second highest enriched motif in p-bodies (see new Fig. S12).

2. 22 references is not a lot, considering that the major impact of this work comes from the experimental setup that should benefit fields such as origin-of-life or modern biology. I wonder whether the authors could better put their results in perspective to what is already known (see above) and highlight better which findings actually were not expected based on previous work. I also think the authors could better outline in the conclusion section, how their experimental setup might solve issues that other groups had so far (eg. By referring to difficulties of in vivo/vitro analysis of condensates, or maybe in respect to synthetic coacervates in the origin of life field..)

We agree with the reviewer that 22 references are probably not sufficient to adequately embed our work in the context of previous work. We have now considerably worked on the text to improve the description of our experiments, to improve the ease of understanding our work and to help the reader to put our work in the context of the literature. We have particularly extended our discussion in the hope of providing a more comprehensive perspective on the literature that our study is based on. Consequently, the number of references now increased from 22 to 45.

3. There are many other parameters (apart from the mentioned potential hybridization) that affect uptake into the droplets. Eg. charge density, ionic strength, amino acid sequence of the main droplet building blocks, viscosity/surface tension,... Now you could argue that, even though neglected here, specific transcripts/motifs were commonly enriched, indicating that these other parameters might be not as important for the uptake of some transcripts. But given the fact that condensates and coacervates might have significant differences in the described parameters (eg. Recent review from Spruijt), a more detailed consideration would help this work.

We believe that reviewer #3 raises a valid point here, as the discussion of our manuscript indeed falls short of a description of the potential influences of other parameters which might affect RNA uptake into coacervates. Therefore, we have incorporated a new paragraph in the discussion (see page 10) which takes other structural and sequence characteristics of RNA as well as RNA independent factors such as ionic strength or peptide sequence into consideration. We believe that our extended discussion now provides a more balanced description of the determinants of RNA uptake into coacervates. Yet, we would also like to point out that we believe that providing the full picture of chemical parameters which determine RNA uptake into coacervates is challenging and probably beyond the scope of this manuscript as there would be enough discussion points to write a full review article on this topic.

Some minor things:

On p4: "In comparison coacervates containing the longest average transcript length were among the smallest coacervates."

On p5: "Our analysis showed that there was no correlation to the transcript length and its frequency in detection into the coacervates"

Not sure if I got it right, but seems contradicting.

This is indeed a misunderstanding. The comment on p4 is discussing the relationship between the coacervate size and the length of the RNA it contains. The comment on p5 discusses if transcript length influences the partition into coacervates in general.

In the SI, the materials part is missing.

We have now incorporated information about all reagents used in the methods section

Generally, some plots (both text and SI) are missing units on the axis

We have now included the missing unit labels (specifically Fig. 2b, S3a, S7a, S9a,c).

S1d is missing, S5b seems to be missing something based on the caption?

We have updated our all Supplementary Figures and thus eliminated the mistakes.

In the SI, when you explain how you prepare the droplets, it would be great if one could better compare the ionic strength and charge density between condensates and coacervates, eg. either express the polycations/anions as mg/mL or maybe better mM of charge for all the types.

We have added the missing information for the polymers. Generally, we would like to thank the reviewer for catching these minor issues with a keen eye. We corrected all mistakes and added the needed information where requested.

REVIEWERS' COMMENTS

Reviewer #1 (Remarks to the Author):

The authors have done a good job of addressing my prior comments. I am supportive of publication in Nature Communications since the work presents a technical achievement of sequencing single condensates, which could be applying in some interesting biological contexts.

Some specific issues to improve the manuscript are below:

1) It would be worthwhile discussing the size constraints on what would be technically feasible in this type of approach.

Sizes are reported in Figure 1, Figure s3, and a size bar in Figure s9d but no units are given to allow the reader to assess the sizes of observed condensates. I suggest:

- a) the authors document in relevant units the sizes of the condensates being sorted and
- b) Discuss what is technically feasible. ie. Could condensates of biologically relevant size be sorted in this manner?

2) It would greatly strengthen the manuscript to test if the motifs they predict for stress granule enrichment are functional by cloning repeated copies of these into a reporter used for stress granule partitioning (e.g. Matheny et al., 2021, RNA). I realize this may be beyond the scope of this work but it is the type of experiment that would demonstrate an important principle (if it had an effect).

Reviewer #2 (Remarks to the Author):

The revised manuscript is technically convincing with the added experiments and would recommend for publication.

One remaining question is that while different types of motifs (mononucleotide repeats) are enriched in murine RNA experiments, I wonder why B1 SINES in murine species were not enriched in this case. Is it due to the difference in their abundance? (Could this be more elaborated in the discussion?) Also, one final suggestion is to include p-values for the enriched motif. This would provide important information to readers with regards to the statistical significance of each motif sequence.

Reviewer #3 (Remarks to the Author):

I appreciate the effort from the authors to revise the manuscript. I believe they have addressed my major concerns and that the manuscript improved notably. I'm looking forward to its final publication in Nature Communications.

REVIEWER COMMENTS

Reviewer #1 (Remarks to the Author):

The authors have done a good job of addressing my prior comments. I am supportive of publication in Nature Communications since the work presents a technical achievement of sequencing single condensates, which could be applying in some interesting biological contexts.

Some specific issues to improve the manuscript are below:

1) It would be worthwhile discussing the size constraints on what would be technically feasible in this type of approach.

Sizes are reported in Figure 1, Figure s3, and a size bar in Figure s9d but no units are given to allow the reader to assess the sizes of observed condensates. I suggest:

- a) the authors document in relevant units the sizes of the condensates being sorted and
- b) Discuss what is technically feasible. ie. Could condensates of biologically relevant size be sorted in this manner?

We thank Reviewer#1 for making us aware that we forgot to specify the size of the scale bars in the microscopy pictures that were added in the as part of the manuscript revision. We have now added this information (Fig. 1, Fig. S1 and Fig. S9).

As for the other figures: We have now added the information that the coacervate size unit is the forward-scatter (FSC) of the FACS wherever we failed to mention it (basically only Fig. S3, in Fig.1 we mentioned that size unit is FSC). We further discuss that condensates found in cells can be sorted, since the limit of condensate sorting is only given by the ability of the FACS machine (p12). (The size limit for particle sorting on a conventional FACS machine are between 300nm and 500nm)

2) It would greatly strengthen the manuscript to test if the motifs they predict for stress granule enrichment are functional by cloning repeated copies of these into a reporter used for stress granule partitioning (e.g. Matheny et al., 2021, RNA). I realize this may be beyond the scope of this work but it is the type of experiment that would demonstrate a important principle (if it had an effect).

We agree that these experiments would be highly interesting but beyond the scope of this manuscript. Still, as we have stated in our previous point-by-point response, the discovered motifs provide an exciting starting point for future studies.

Reviewer #2 (Remarks to the Author):

The revised manuscript is technically convincing with the added experiments and would recommend for publication.

One remaining question is that while different types of motifs (mononucleotide repeats) are enriched in murine RNA experiments, I wonder why B1 SINES in murine species were not enriched in this case. Is it due to the difference in their abundance? (Could this be more elaborated in the discussion?) Also, one final suggestion is to include p-values for the enriched motif. This would provide important information to readers with regards to the statistical significance of each motif sequence.

Reviewer #2 raises an interesting question which we haven't considered. Indeed, if one looks closely at the discovered motifs (Suppl. Fig. 11) there is a single motif (motif#5) which stands out because it is clearly not a mononucleotide repeat. Interestingly, we found that this motif has almost perfect complementarity to the B1 SINES as annotated by RepBase. Hence, we added a new Suppl. Fig. 11c to demonstrate this important finding and would like to thank Reviewer#2 for the suggestion to explore B1 SINES.

We further provide the E-values (equal to the combined p-value of the sequence times the number of sequences in the database) which informs about how significantly the motifs are enriched for in Suppl. Fig. 11 (as well as Suppl. Fig. 12 which was also missing E-values).

Reviewer #3 (Remarks to the Author):

I appreciate the effort from the authors to revise the manuscript. I believe they have addressed my major concerns and that the manuscript improved notably. I'm looking forward to its final publication in Nature Communications.

We would like to thank the reviewer for his/her efforts in improving our manuscript.